# AdaFace – A Versatile Face Encoder for Zero-Shot Diffusion Model Personalization

Reference "Jedi fighting pose with a lightsaber"    Reference "Dancing pose among folks in a park"

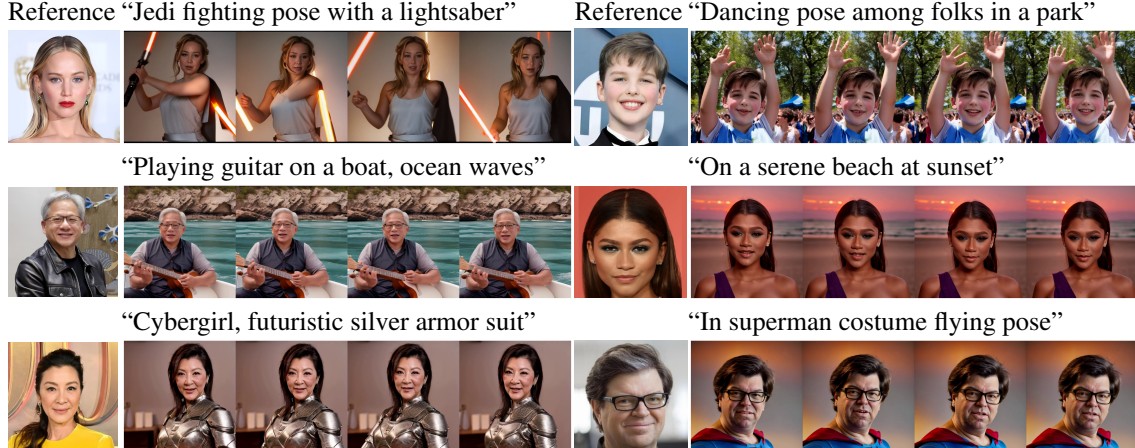

"Playing guitar on a boat, ocean waves"    "On a serene beach at sunset"

"Cybergirl, futuristic silver armor suit"    "In superman costume flying pose"

Figure 1: Although AdaFace is solely trained on static images, the subject embeddings it generates can directly condition AnimateDiff to produce personalized videos across diverse scenes without requiring any modifications.

## Abstract

Since the advent of diffusion models, personalizing these models – conditioning them to render novel subjects – has been widely studied. Recently, several methods propose training a dedicated image encoder on a large variety of subject images. This encoder maps the images to identity embeddings (ID embeddings). During inference, these ID embeddings, combined with conventional prompts, condition a diffusion model to generate new images of the subject. However, such methods often face challenges in achieving a good balance between authenticity and compositionality – accurately capturing the subject's likeness while effectively integrating them into varied and complex scenes. A primary source for this issue is that the ID embeddings reside in the *image token space* ("image prompts"), which is not fully composable with the text prompt encoded by the CLIP text encoder. In this work, we present AdaFace, an image encoder that maps human faces into the *text prompt space*. After being trained only on 400K face images with 2 GPUs, it achieves high authenticity of the generated subjects and high compositionality with various text prompts. In addition, as the ID embeddings are integrated in a normal text prompt, it is highly compatible with existing pipelines and can be used without modification to generate authentic videos. We showcase the generated images and videos of celebrities under various compositional prompts. The source code is released on an anonymous repository `https://github.com/adaface-neurips/adaface`.

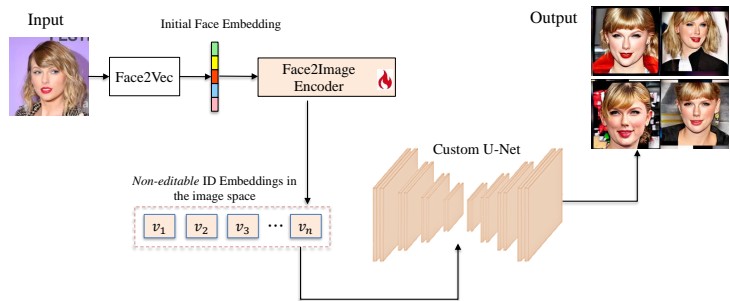

Figure 2: A typical zero-shot face encoder pipeline for diffusion models. First, a Face2Vec module (e.g., ArcFace [Deng et al., 2019]) extracts a single vector that captures the facial features. Then a trainable Face2Image encoder (e.g., Arc2Face [Papantoniou et al., 2024]) maps it to $n$ facial tokens $v_1, \cdots, v_n$ within the image embedding spaces. The facial tokens condition the U-Net (either original or fine-tuned) to generate authentic-looking subject images. However, since the facial tokens is not blended with other text prompts (sometimes they are simply concatenated), the whole pipeline has weaker compositionality than using text prompts alone. Moreover, such models are often incompatible with existing diffusion pipelines, such as AnimateDiff Guo et al. [2024a].

## 1 Introduction

Recent years have witnessed the blossom of diffusion models, which have been widely used in image generation, image editing, and video generation [Ho et al., 2020, Nichol et al., 2022, Saharia et al., 2022, Rombach et al., 2022, Podell et al., 2024, Chen et al., 2024a, Kawar et al., 2023, Peebles and Xie, 2023, Guo et al., 2024a]. A particularly interesting application of these models is personalization, where they are conditioned to generate images of specific subjects. Previously, this was primarily achieved through test-time fine-tuning [Ruiz et al., 2022, Gal et al., 2022a, Kumari et al., 2022, Tewel et al., 2023], which introduced additional computational demands and complexity to the image generation process. Recent advancements have seen the development of zero-shot, tuning-free methods [Wei et al., 2023, Ye et al., 2023, Shi et al., 2023, Wang et al., 2024, Papantoniou et al., 2024, Guo et al., 2024b, Huang et al., 2024, Han et al., 2024, Chen et al., 2024b, He et al., 2024]. These methods train a dedicated image encoder to convert subject images to identity embeddings (ID embeddings) using a large dataset. During inference, these ID embeddings are combined with standard text prompts to generate new images of the subject (Figure 2). Despite these innovations, these approaches often struggle to strike a good balance between authenticity and compositionality. Authenticity ensures the model captures the true likeness of the subject, whereas compositionality concerns the model's ability to seamlessly integrate the subject into diverse and intricate scenes. The challenge primarily stems from how ID embeddings are utilized: in many zero-shot methods, the embeddings exist in the *image token space* ("image prompts") and do not fully mesh with text prompts. In cases like [Huang et al., 2024], while the ID embeddings are within the text space, there lacks targeted training to enhance their integration with other text prompts, resulting in compromised compositionality.

Given the limitations of existing methods, we propose AdaFace, a versatile face encoder that maps human faces into the *text prompt space*. First, the ID embeddings generated by AdaFace seamlessly integrate with text prompts via the CLIP text encoder, allowing for more coherent and expressive conditioning. Second, we employ targeted training strategies to enhance the compositionality of the ID embeddings, ensuring they are able to be used to generate diverse and complex scenes. Furthermore, AdaFace is highly compatible with existing diffusion pipelines, requiring no modifications to generate authentic videos, as demonstrated in Figure 1. Notably, due to efficient model design and distillation techniques, AdaFace is trained on merely 406,567 face images with 2 RTX A6000 GPUs, all within a constrained compute budget.

We demonstrate the effectiveness of AdaFace by showcasing the generated images and videos of celebrities under various compositional prompts. We also perform quantitative evaluations to validate that AdaFace achieves a good balance between authenticity and compositionality, measured by ArcFace similarity and CLIP-Text similarity, respectively.

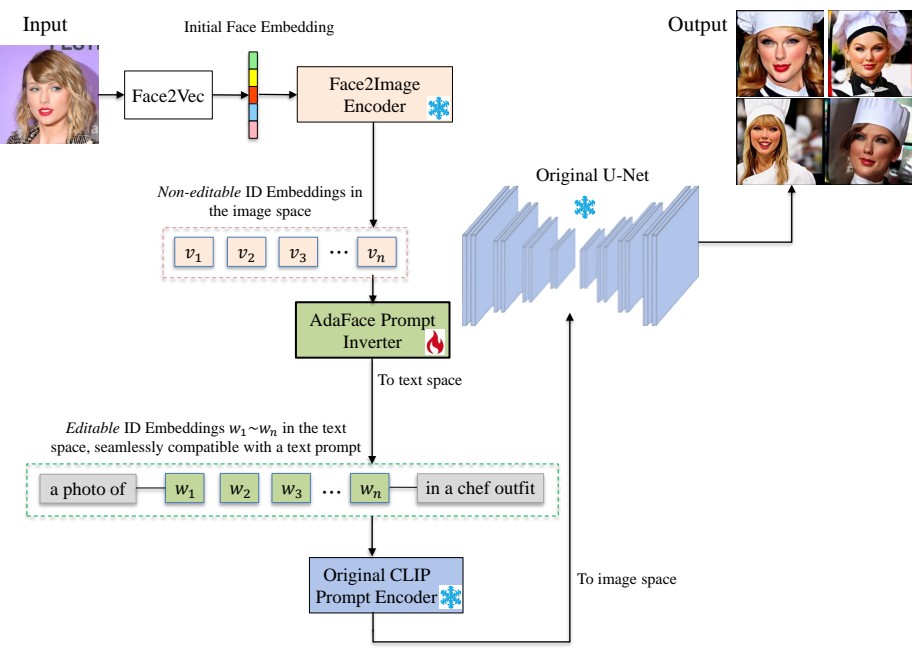

Figure 3: The core of AdaFace is the *Prompt Inverter*, which inverts the image-space ID embeddings from another model to the *text prompt space*, represented as $w_1, \cdots, w_n$. These embeddings are integrated into a standard text prompt and encoded by a CLIP prompt encoder. CLIP coherently composes the semantics of the ID embeddings and the text prompt, providing good compositionality.

## 2 Method

Motivated by the advantages of *text space* face prompts, we propose techniques to distill one or more image-space face encoders into the text space, and further enhance its compositionality. The overall architecture of AdaFace is shown in Figure 3. The core module of AdaFace is the *AdaFace Prompt Inverter*, which inverts the image-space ID embeddings to the text space, enabling the integration of the ID embeddings into a standard text prompt. The ID embeddings are then encoded by a CLIP prompt encoder, which coherently composes the semantics of the ID embeddings and the text prompt. The text-level composition also facilitates *Composition Distillation* (Figure 5), which significantly improves the compositionality of the ID embeddings without additional training data. A side-effect of composition distillation is that, when there is spatial misalignment between the subject-single and subject-composition images, the subject features will be gradually contaminated by background features, reducing their authenticity. Accordingly, we propose a *Elastic Face Preserving Loss* (Figure 6), to prevent the subject features from degeneration.

### 2.1 AdaFace Architecture

The core module of AdaFace is the *AdaFace Prompt Inverter*, which converts the image-space ID embeddings from a Face2Image model to the text space.

The architecture and initialization of the prompt inverter significantly impacts the training efficiency. Compared to other deep learning tasks, the diffusion training is highly stochastic and the gradients have a much lower signal-to-noise ratio. It is highly challenging to train a sizable diffusion component from scratch without high compute budgets and large batch sizes. To achieve efficient learning, we adopt the same architecture as the CLIP text encoder for the AdaFace Prompt Inverter, and initialize it with the pre-trained weights. This ensures that the output embeddings are not very distant from the text space from the beginning of training, and the model learns more signals from the gradients.

One may raise the question that since the output of a pre-trained CLIP encoder is in the image space, why it is able to adapt quickly to generate text-space embeddings? We speculate that in CLIP, the semantics of low-level layers and high-level layers are not in totally incompatible spaces, but rather,

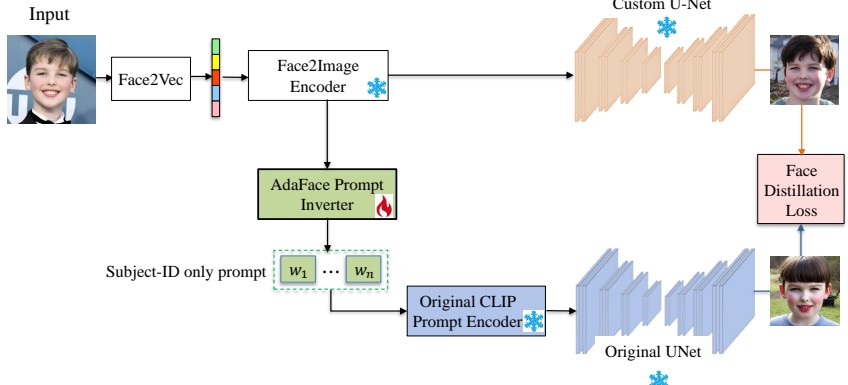

Figure 4: Face distillation on face images. The output of the AdaFace stream is compared with the Face2Image stream. During this process, only the AdaFace Prompt Inverter is optimized.

the high-level semantics enrich the low-level ones. Our hypothesis is corroborated by [Toker et al., 2024], as well as the community practice of ad-hoc fusing the output embeddings of multiple CLIP text encoder layers[1]. The semantics of layer features gradually transition from the text space to the image space. As a result, during fine-tuning, the skip connections within CLIP will allow the low-level semantics to take shortcut towards the output embeddings, and the high-level layers will gradually learn to enrich the low-level semantics in the *text space* instead.

The training of the prompt inverter is divided into two stages. In the first *face distillation* stage, a Face2Image model guides the prompt inverter to generate authentic faces in the text prompt space. In the second *composition distillation* stage, the prompt inverter observes how the original model output responds to compositional prompts, and learns to generate similar responses, so as to allow the text prompts to control the composition of the generated images.

## 2.2 Face Distillation

The face distillation stage is illustrated in Figure 4, where the objective is to minimize the difference between the generated images by the original Face2Image model and by the AdaFace Prompt Inverter on the same initial noise. The training objective, namely the face distillation loss, is formulated as a reconstruction loss between the two generated images:

$$\mathcal{L}_{\text{face}} = \mathbb{E}_{f \sim F, z \sim \mathcal{N}(0,I), t \in [1,T]} \left[ \|G_{\text{AdaFace}}(f,z,t|\theta) - G_{\text{Face2Image}}(f,z,t|\theta')\|_2^2 \right], \quad (1)$$

where $G_{\text{Face2Image}}$ and $G_{\text{AdaFace}}$ are the Face2Image and the AdaFace Prompt Inverter conditioned U-Nets, respectively, $f$ is a random face drawn from the face space $F$, $z$ is the initial noise, and $\theta$ and $\theta'$ are the parameters of the AdaFace Prompt Inverter and the Face2Image model, respectively. For some models such as Ada2Face, $\theta' \neq \theta$.

In order to sweep the input space $\{f, z, t\}$ as completely as possible, we adopt a few techniques:

**Random Gaussian Face Embeddings.** Empirically, we observe that almost all random face embeddings result in legitimate face images when processed by the Face2Image model. Therefore, we expand the candidate face space $F$ by including random face embeddings drawn from a Gaussian distribution, alongside the face embeddings extracted from real face images: $F = F_{\text{real}} \cup F_{\text{rand}}$.

**Multi-Timestep Distillation.** We use multiple denoising steps on the same initial noise, and compute the reconstruction loss on all the steps, so that the prompt inverter learns to imitate the Face2Image model's behavior on intermediate noise levels:

$$\mathcal{L}_{\text{face}} = \mathbb{E}_{f \sim F, z_1 \sim \mathcal{N}(0,I), t_1 > \cdots > t_k \in [1,T]} \sum_{i=1}^{k} \left[ \|G_{\text{AdaFace}}(f, z_i, t_i|\theta) - G_{\text{Face2Image}}(f, z_i, t_i|\theta')\|_2^2 \right], \quad (2)$$

---

[1]https://github.com/AUTOMATIC1111/stable-diffusion-webui/discussions/5674

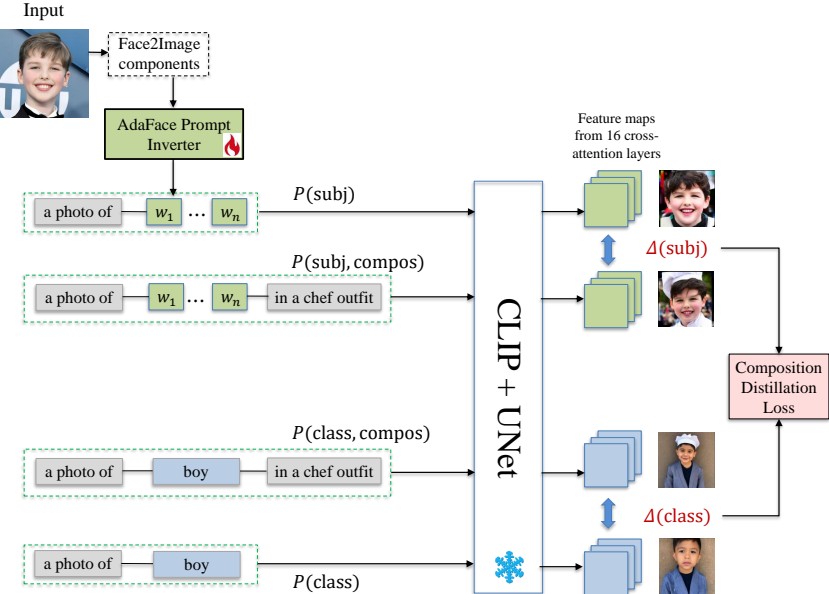

Figure 5: Composition distillation on four types of prompts: subject-single, subject-composition, class-single and class-composition. The four generated images form two contrastive pairs, and their feature deltas are encouraged to be aligned through a composition distillation loss.

where $t_1, \cdots, t_k$ are a randomly sampled sequence of timesteps, and when $i > 1$, $z_i$ is the partially denoised image by $G_{\text{Face2Image}}$ in the previous step.

**Dynamic Model Expansion.** When the training loss plateaus, it suggests that the model has reached the limits of its capacity to capture nuanced facial features. In this situation, we expand the model capacity by incorporating additional *query* and *value* projections within the attention layers of the prompt inverter. As a result, each token is represented by multiple, subtly distinct query and value tokens. This enables the model to better grasp the subtle facial features of the subject, thanks to the increased diversity and richness of the queries and values. Note that the number of keys and output tokens remain unchanged, ensuring that the computational load does not increase drastically.

Specifically, when a query projection $Q$ is expanded by $N$ times, we make $N$ identical copies of $Q$ and add Gaussian noises to $N - 1$ of them. The same operation is applied to the value projection $V$. This is to ensure that the expanded $Q'$ and $V'$ do not deviate too much from the original $Q$ and $V$, and the model augments the original features with slightly varied replicas.

The attention expansion proves to be particularly beneficial at the lower layers of the prompt inverter. Intuitively, once some information in the features from the upstream Face2Image encoder is lost in the lower layers, it is hard to recover in the higher layers. The mechanism of expanding queries and values creates multiple, slightly varied replicas of the same information, thereby allowing the model to select the most informative copy for preservation and further processing in subsequent layers. This approach is conceptually akin to the role of the excitation operator in a squeeze-and-excitation network [Hu et al., 2018], which also emphasizes selectively retaining the most significant features.

## 2.3 Composition Distillation

A prevalent issue with existing face encoders is that the subject token tends to dominate the generated images, resulting in degeneration of compositionality. To mitigate this issue, we employ *composition distillation* (Figure 5) to regularize the subject embeddings, ensuring that their semantics are effectively integrated with other tokens, enhancing the overall expression. During this process, the model observes how the original diffusion model adjusts output features to incorporate additional compositional prompts into the output image. The model then imitates these adjustments when encountering similar compositional prompts.

For this purpose, four types of prompts are employed to form two contrastive pairs: 1) a "subject-single" prompt that only contains the subject, such as "A photo of a [Zendaya]", 2) a "subject-composition" prompt such as "A photo of a [Zendaya] in the forest", 3) a "class-single" prompt that only contains a general class, such as "A photo of a woman", and 4) a "class-composition" prompt such as "A photo of a woman in the forest". Ideally, the semantic differences between "A photo of $x$" and "A photo of $x$ in the forest" should only be relevant to "in the forest", and is independent of $x$.

We represent the semantic differences between two pairs of prompts as their "feature deltas". The training objective is to encourage the feature deltas between the subject-single and subject-composition images to be aligned with the feature deltas between the class-single and class-composition images. In other words, the following equation is expected to hold approximately:

$$\Delta(\text{subject, compos}) \doteq \text{feat(subject, compos)} - \text{feat(subject)}$$
$$\approx \Delta(\text{class, compos}) \quad \doteq \text{feat(class, compos)} \quad - \text{feat(class)}, \tag{3}$$

where subject, class, (subject, compos) and (class, compos) denote the four types of prompts, respectively. (subject, compos) and (class, compos) are randomly drawn from a pool of common compositional prompts consisting of various backgrounds, additional objects, dresses, image styles and lighting conditions. $\text{feat}(x)$ refers to relevant features, including 1) the output features from all the cross-attention layers, 2) the attention maps in all the cross-attention layers, and 3) the encoded prompt embeddings by CLIP text encoder. $\text{feat}(x) - \text{feat}(y)$ is the *orthogonal subtraction* between two feature maps, defined below.

We define a *compositional delta loss* that aligns the feature deltas $\Delta_i(\text{subject, compos})$ and $\Delta_i(\text{class, compos})$ on the three types of features listed above:

$$\mathcal{L}_\Delta = \sum_i \{1 - \mathbb{E}_{\text{compos}\sim U(C)} \cos(\Delta_i(\text{subject, compos}), \Delta_i(\text{class, compos}))\}, \tag{4}$$

in which $i$ indexes the feature type (cross-attention output features, attention maps or CLIP prompt embeddings), and $U(C)$ is a uniform distribution on a set of compositional prompts $C$.

**Orthogonal Subtraction.** We wish to remove subject-specific features through the feature subtraction "$\text{feat(subject, compos)} - \text{feat(subject)}$". However, it is commonly observed that the subject-specific features may have different magnitudes (often smaller under compositional prompts). To mitigate this issue, we propose to use orthogonal subtraction, which is invariant to the scale of the subject-specific features. A relevant idea [Wang et al., 2023] is explored for language model fine-tuning. Specifically, the feature deltas are calculated using the following equation:

$$\Delta\text{feat}(s, c) = \text{feat}(s, c) - \text{proj}_{\text{feat}(s)}(\text{feat}(s, c)), \tag{5}$$

where $\text{proj}_{\text{feat}(s)}(\text{feat}(s, c))$ is the projection of $\text{feat}(s, c)$ onto $\text{feat}(s)$, computed as:

$$\text{proj}_{\text{feat}(s)}(\text{feat}(s, c)) = \langle \text{feat}(s, c), \text{feat}(s) \rangle \text{feat}(s), \tag{6}$$

with $\langle \text{feat}(s, c), \text{feat}(s) \rangle$ being the inner product between the two features. The operation effectively projects $\text{feat}(s, c)$ onto the orthogonal complement of $\text{feat}(s)$ and then subtracts this projection from $\text{feat}(s, c)$. As a result, $\Delta\text{feat}(s, c)$, the feature delta, is orthogonal to $\text{feat}(s)$. This methodology ensures that the deltas remove as much of the subject-specific features as possible, thereby minimizing the influence of the scales of the subject-specific features contained within $\text{feat}(s, c)$.

**Differences with Previous Methods.** While previous methods have explored analogous concepts, such as StyleGAN-NADA [Gal et al., 2022b], which applies similar regularizations in the CLIP prompt embedding space, and PuLID [Guo et al., 2024b], which introduces similar contrastive regularizations on cross-attention queries, our approach is more comprehensive and effective. Our compositional delta loss encompasses a broader range of relevant features, including the attention maps and output features from cross-attention layers, and the CLIP prompt embeddings. Moreover, we introduce an orthogonal subtraction technique for computing the feature deltas. This technique isolates and extracts composition-specific features, making the distillation more effective.

## 2.4 Elastic Face Preserving Loss

The composition distillation is done on instances with different prompts starting from the same initial noise. This is to encourage the diffusion model to generate images that are compositionally similar

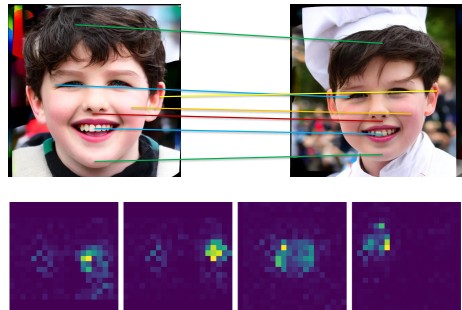

Figure 6: To prevent subject features from degeneration due to spatial misalignment during composition distillation, we propose a Elastic Face Preserving Loss. The second row shows the cross-attention maps at selected four points on the subject-single image. The highlighted pixels associate the corresponding facial areas across the two images. The features of matching pixels are required to be close to each other to achieve subject feature preservation.

[Zhang et al., 2024], to achieve more accurate alignment between the image pairs. Despite this effort, spatial misalignment often persists between the images differently prompted. This misalignment can result in delta loss providing erroneous signals from non-facial to facial areas, slowly reducing the authenticity of the generated subjects. For instance, on a noisy input face image, the output image from the subject-single instance is expected to largely retain the same facial contours as the input. However, the output from the subject-composition instance often deviate from the original face contours, due to the introduction of additional compositional elements. An illustrative example provided in the first row of Figure 6 shows how a chef hat in one image spatially aligns with the hair in another, leading to potential contamination in the subject's hair representations.

To tackle this challenge, we view the subject-composition image as a "warped" version of the subject-single image, and turn to techniques from the Optical Flow literature[Teed and Deng, 2020, Sui et al., 2022] to estimate a matching field. The matching field is used to spatially align the subject features across different images, ensuring them to be consistently maintained after "warping".

Specifically, the model takes as input a noisy face image from the training data. The face image is accompanied by a segmentation mask, isolating the face area for matching. We compute the cross attention matrix[2] between the queries of a subject-single instance and a subject-composition instance:

$$\text{CA}(\text{subj}, \text{compos}) = \text{softmax}(Q_{\text{subj}} Q_{\text{compos}}^T), \tag{7}$$

By looking up the cross-attention map $\text{CA}(\text{subj}, \text{compos})$, we can find the pixels best matching a subject-single image pixel in a subject-composition image. The second row in Figure 6 shows the attention maps of four points on the face in the left image. We "soft-warp" the subject-composition features to align with the subject-single features through matrix multiplication, and require the warped features to be close to the facial features in the subject-single image:

$$\mathcal{L}_{\text{face-preserving}} = 1 - \cos\Big(\text{CA}(\text{subj}, \text{compos}) \odot \text{feat}(\text{compos}), \text{feat}(\text{subj})\Big)_{\text{mask}}. \tag{8}$$

Here for clarity, feat(subject, compos) is abbreviated as feat(compos). The cosine similarity $\cos(\cdot, \cdot)$ is computed on the masked area. The face-preserving loss is computed on each U-Net cross-attention layer. It encourages the subject features in the subject-composition instance to be consistent with those in the subject-single instance, preventing them from being contaminated in the composition distillation process.

---

[2]The inner product is not scaled to make the matching scores more polarized.

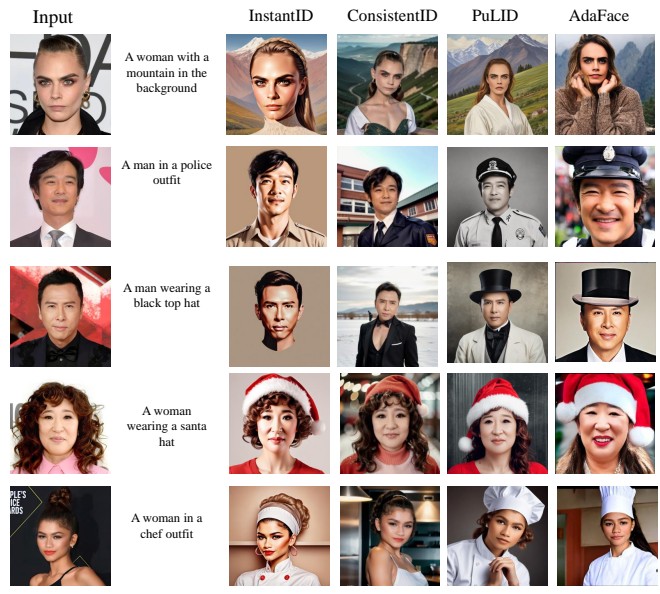

Figure 7: Qualitative comparison of AdaFace with state-of-the-art face encoders. AdaFace generates images that maintain the highest authenticity of the subjects, while still follow the target prompts.

# 3 Experiments

## 3.1 Dataset and Training Details

We trained AdaFace on a combination of two face datasets: Flickr-Faces-HQ (FFHQ) [Karras et al., 2019], which comprises 70,000 images, and VGGFace2-HQ [Cao et al., 2018], which comprises 336,567 images after filtering. Face masks were generated using the BiSeNet face segmentation model [Yu et al., 2018]. The distilled Face2Image model is Ada2Face [Papantoniou et al., 2024], as it is able to generate authentic and diverse face images. The training employed the Prodigy optimizer [Mishchenko and Defazio, 2024] with d_coef=2 (akin to the learning rate in other optimizers) during face distillation, and d_coef=0.5 during composition distillation. Batch sizes were set to 4 and 3 for the two stages, respectively, with a gradient accumulation of 2. The model was trained with 240,000 iterations in the face distillation stage and 120,000 iterations in the composition distillation stage. During face distillation, the loss reached a plateau twice, resulting in two dynamic expansions of the model capacity. Eventually, the attention layers in the trained prompt inverter were expanded with multipliers of $(8x, 8x, 8x, 4x, 4x, ..., 4x)$ relative to the original CLIP text encoder. This resulted in a total of 2M parameters, in contrast to the 1.2M parameters of the original model.

In addition, we collected the images of 23 celebrities, each with 9 10 images, as the evaluated subjects. These celebrities include actors, singers and internet celebrities on Instagram. This dataset will be released along with the code.

## 3.2 Qualitative Comparisons

We compared AdaFace with a few state-of-the-art face encoders, including InstantID [Wang et al., 2024], ConsistentID [Huang et al., 2024] and PuLID [Guo et al., 2024b]. The input were images from our celebrity-23 dataset.

The results presented in Figure 7 demonstrate that AdaFace produces images that not only exhibit high authenticity of the subjects but also show good consistency with the text prompts. In comparison, other models often fall short in generating images that are either less authentic or less compositional. For instance, InstantID tends to produce overly stylized images with significant variability in authenticity across different subjects. PuLID, while generating aesthetically pleasing images, achieves slightly lower authenticity levels compared to AdaFace. Despite also utilizing a text-space approach,


Yann Lecun in a white apron and chef hat,
garnishing a gourmet dish

Jensen Huang dancing pose among folks in a
park, waving hands


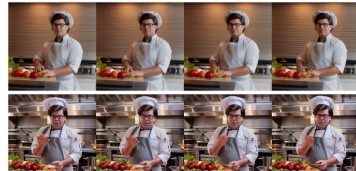 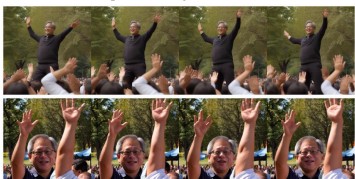

Figure 8: Comparison of AdaFace with ID-Animator on personalized video generation. AdaFace generates videos with higher authenticity and compositionality.

ConsistentID has the least compositional output among the models evaluated, largely due to the absence of compositional training in its ID embeddings.

In addition, we plugged AdaFace into AnimateDiff, and generated personalized videos of celebrities under various compositional prompts. The results are shown in Figure 1. Figure 8 compares with a recent method ID-Animator [He et al., 2024]. AdaFace generated videos with high authenticity and compositionality, while ID-Animator usually produces videos with less authentic subjects.

## 3.3 Quantitative Evaluations

To assess the performance of AdaFace quantitatively, we evaluated a few baseline methods and AdaFace, on the "celebrity-23" images and DreamBench compositional prompts, comparing AdaFace with two baseline methods PuLID and InstantID. First, we measured the face similarity using the cosine similarity between the ArcFace embedding of the generated images and reference images. In addition, the CLIP-Text (CLIP-T) metric determines the consistency of the generated images with the prompts. The DINO and CLIP-I metrics are less indicative and are only for reference. The results, detailed in Table 1, show that AdaFace achieved comparable face similarity and prompt consistency scores to PuLID, and slightly outperformed InstantID. Note that the results of AdaFace is achieved on the original Stable Diffusion 1.5 model weight, which usually leads to much lower composition scores than other fine-tuned SD 1.5 model weights, such as RealisticVision.

| | **ArcFace (subj)** | **CLIP-T (comp)** | DINO | CLIP-I |
|---|---|---|---|---|
| DB | 0.349 | **0.324** | 0.470 | 0.656 |
| TI | 0.326 | 0.250 | 0.508 | 0.675 |
| PuLID | 0.468 | **0.280** | 0.512 | 0.630 |
| InstantID | 0.455 | 0.257 | 0.472 | 0.595 |
| Ada | **0.476** | 0.270 | 0.544 | 0.670 |
| -Comp | 0.505 | 0.235 | 0.598 | 0.685 |

Table 1: Quantitative evaluation on the "celebrity-23" images and DreamBench compositional prompts. **-Comp** is the model trained only with the face distillation stage.

As an ablation study, we list the performance of the AdaFace model without composition distillation. It can be seen that the face authenticity is slightly reduced after composition distillation, however, the generated images become much more consistent with the prompts.

## 4 Conclusions and Discussions

In this work, we present AdaFace, a versatile face encoder that maps human faces into the text prompt space. AdaFace is trained with a low compute budget and achieves high authenticity and compositionality in zero-shot generation of subject images. We demonstrate the effectiveness of AdaFace by showcasing the generated images and videos of celebrities under various compositional prompts. Additionally, our quantitative evaluations further underscore its performance.

A notable limitation of AdaFace is that the authenticity of the output face embeddings are constrained by the Face2Image model it distills from. However, this limitation can be addressed by distilling on more powerful Face2Image models and expanding the model capacity. For future work, we would extend the AdaFace method to object images. For instance, applying AdaFace distillation techniques to IP-Adapter [Ye et al., 2023] could enable the generation of both human and object images.

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
