# OpenReview forum: "AdaFace: A Versatile Face Encoder for Zero-Shot Diffusion Model Personalization"
_NeurIPS.cc/2024/Conference — Submitted to NeurIPS 2024_

### Official Review · Reviewer_jpRS · 2024-07-10

**Soundness:** 1
**Presentation:** 1
**Contribution:** 2
**Rating:** 3
**Confidence:** 5

**Summary:**

This work presents a zero-shot face-generation method based on diffusion models. The proposed method first extracts face features using Face2Vec and trains a network to map these features into the textual space (i.e., the prompt embedding space for diffusion’s text condition). The main difference from existing zero-shot face generation approaches is that, instead of adding conditions in the denoising UNet, the presented method incorporates the condition into the textual embedding. To prevent the subject information from overwhelming the generation (e.g., preserving only the subject ID while ignoring other descriptions), the authors propose a Composition Distillation Loss. This contrastive loss encourages the model to generate descriptions other than the subject information.

The main shortcoming of the paper lies in the experimental validation. There are multiple components proposed, but their effectiveness is not adequately demonstrated. Additionally, both qualitative and quantitative evidence fail to show that the proposed method outperforms SoTA methods.

**Strengths:**

* The Compositional Distillation Loss is an interesting and intuitive approach to reducing the problem of subject information overwhelming the generation

**Weaknesses:**

* There is a lack of intuition behind using multi-timestep distillation. This approach can cause accumulated errors, and there is no evidence provided to support the benefits of adopting such a strategy.
* The section on dynamic model expansion is very unclear. It is not specified which part of the model is expanded or if the tokens are simply replicated. While adding Gaussian noise to replicated tokens is mentioned, there is no empirical validation of its effectiveness.
* The qualitative comparison, especially in Figure 7, does not support the claim that the proposed method has advantages over baselines like PuLID. Additionally, the benchmarking results in Table 1 show limited improvement in facial identity preservation, and the text alignment can be inferior to PuLID. Thus, the experiments are not comprehensive enough to convincingly demonstrate that the proposed approach is superior to existing methods like PuLID.

**Questions:**

My suggestions to the author are follows:
* Conduct a comprehensive ablation study of the components proposed in the paper, which could include but is not limited to: 1) Justification for placing the learnable part (the conditioning module) in the text encoder rather than the diffusion denoising UNet. 2) Effectiveness of using facial preservation loss, multi-timestep distillation, and dynamic model expansion.

* Design experiments that are more relevant to the proposed methods. The current paper contains many irrelevant experiments/visualizations, such as: Figure 1/8: Applying AnimateDiff and generating video. Figure 2: Illustrating other work's pipeline.
Figure 6: Feature alignment lines and heatmap that do not provide useful information. The compatibility of the proposed method with different backbones/plugins can be mentioned in the paper but should not be the main focus and can be presented in the supplementary materials.

* Polish the language and improve readability (e.g., using tools like ChatGPT). Some content is not presented in a formal paper format, such as Line 222, "each with 9 10 images."

**Limitations:**

The authors have discussed the limitations.

---

> ### Author Rebuttal · Authors · 2024-08-07
>
> Thank you reviewer jpRS for your thorough and detailed review of our paper. We appreciate the time and effort you have invested in providing your critical feedback. While some of your comments are indeed challenging, we believe they are valuable for improving the quality and rigor of our work.
>
> 1\. **Multi-timestep distillation**.
>
> While it's true that the student model accumulates larger errors with more steps, the effect of aligning the output with the teacher model will *reduce* accumulated errors. This is because we assume the teacher model is the gold standard at any number of timesteps. Therefore, aligning the student output with any timesteps will help reduce the intermediate errors in all steps. Similar techniques have been adopted in other topics, such as optical flow estimation (Section 3.3 of [a]).
>
> 2\. **Dynamic model expansion**.
>
> The dynamic model expansion involves duplicating the Q (query) and V (value) projection *layers* (as opposed to *tokens*) in the attention layers. The weights of the extra copies of Q, V projections are perturbed with Gaussian noises to introduce variability. We know that each vanilla attention layer has only one Q/K/V projection. After expansion by a factor of $N$, there are $N$ Q,V projections and 1 K projection. Consequently, each input token is mapped to $N$ queries and values, but only 1 key. The number of output tokens is determined by the number of keys, and therefore, the number of output tokens does not change.
>
> 3\. **Relevance to video generation**.
>
> We respectfully disagree with the notion that video generation is not relevant to our topic of subject-driven image generation. As the community shifts focus from image generation to video generation, we believe that the ultimate goal of subject-driven generation is to develop general methods applicable to both images and videos.
>
> A long-standing issue with current video generation models, such as Gen3 and Luma Dream Machine, is that subject consistency deteriorates as the video becomes longer. Dedicated subject embeddings could serve as a long-term memory to maintain subject consistency in very long videos. As a preliminary exploration, we hope our method may inspire more research works along this direction.
>
> Due to the limitations of AnimateDiff, we are currently only able to showcase our method on 1-second videos. However, we aim to demonstrate the broad potential of our method of generating longer videos with more powerful open-source pipelines.
>
> 4\. **Comparison with PuLID**.
>
> We clarify that we do not claim superior performance compared to PuLID. Instead, our focus is to provide the community with a face encoder that performs comparably to existing SOTA methods, with the unique advantage of being plug-and-play with most existing pipelines, such as animate generation. Specifically, inherited from Arc2Face, AdaFace is capable of generating highly realistic images. In contrast, existing SOTA methods, PuLID included, tend to produce more artistic images, as demonstrated in Figure 7 and the attached figure PDF.
>
> To gain an intuitive sense of how our model performs compared to others, we invite you to try our online demo hosted on Huggingface.
>
> 5\. **Ablation study of the proposed components.**
>
> We acknowledge the absence of some ablation studies. Due to limited response time and our small team's low computational resources, it has been unrealistic to train ablated models within this short period. We will add these ablated results to a newer version of the paper as soon as they are available.
>
>
> [a] RAFT - Recurrent All-Pairs Field Transforms for Optical Flow, ECCV 2020.

---

> > ### Comment · Reviewer_jpRS · 2024-08-11
> >
> > The reviewer thanks author's feedback.
> >
> > The reviewer finds the motivation of "Multi-tiemstep distillation" and "Dynamic model expansion" in author's feedback . However, for a top tier conference like NeurIPS, the reviewer believes the work needs to be 1) technically sound by having justifications for proposed components/ideas and 2) experimentally show superior performance than existing works or comparable performance under a harder scenario.
> >
> > Specifically, the reviewer thinks the author has not well justified their proposed methods (see Questions 1) and has not shown progess in the personalized facial generation. As such, the reviewer decides to remain my rating.

---

> > > ### Author Response · Authors · 2024-08-11
> > > **Thanks for your response**
> > >
> > > Thank you for your feedback and for taking the time to review our work.
> > >
> > > We appreciate your recognition of the motivations behind "Multi-timestep Distillation" and "Dynamic Model Expansion." However, we respectfully disagree with your assessment that our work "has not shown progress in personalized facial generation."
> > >
> > > In our submission, we have demonstrated advancements in personalized facial generation, particularly through seamless integration with the existing video generation pipeline, AnimateDiff without fine-tuning. This plug-and-play ability is a unique advantage of our method. Our empirical results, as presented in the paper, show that our approach can generate facial images comparable to several state-of-the-art methods. Moreover, when integrated with AnimateDiff, it demonstrates significant advantages over the existing subject-driven video generation method, ID-Animator.
> > >
> > > While we understand that our contributions may not have been fully aligned with the specific criteria you were looking for, we believe that our work offers meaningful advancements that contribute to the broader field of personalized image and video generation.
> > >
> > > We respect your decision to maintain your rating and will take your feedback into account as we continue to refine and improve our work. Your insights have been valuable and we are committed to addressing the areas you highlighted in future revisions.
> > >
> > > Thank you once again for your careful consideration.

---

> > > > ### Comment · Reviewer_jpRS · 2024-08-13
> > > >
> > > > The reviewer thanks the authors for their response but would like to point out that the claim of "seamless integration with AnimateDiff without fine-tuning" as a contribution of this work is difficult to acknowledge based on the current version of the paper.
> > > >
> > > > If one examines the demos provided by AnimateDiff in their official GitHub repository or GitHub page, it becomes apparent that many works can achieve "seamless integration with AnimateDiff without fine-tuning," such as LoRA or Dreambooth. Even less well-defined conditions, such as layout-based methods (e.g., https://github.com/frank-xwang/InstanceDiffusion/tree/main?tab=readme-ov-file#third-party-implementations), can achieve similar integration. The common factor among these approaches is that they all share the same backbone model as AnimateDiff.
> > > >
> > > > Therefore, the ability to integrate seamlessly with AnimateDiff without fine-tuning seems to be a characteristic of diffusion models that share the same backbone as AnimateDiff, rather than a unique feature of the proposed method. As a result, the reviewer does not agree with the authors' assertion that AnimateDiff integration should be considered a significant contribution of this work.

---

> ### Author Response · Authors · 2024-08-13
> **Thanks for your interesting question**
>
> Thank you for taking the time to carefully examine our claim of "seamless integration with video generation pipelines." We appreciate the opportunity to further clarify and differentiate our approach from similar techniques.
>
> 1. While methods like DreamBooth, LoRA, and more recently, MagicMe [a], can achieve subject consistency in video generation, these approaches require cumbersome subject-specific fine-tuning, which limits their practicality for long videos featuring multiple subjects. These limitations have partly driven the development of recent zero-shot subject-driven generation methods.
>
> 2. More importantly, subject-driven video generation, often referred to as **visual story generation**, aims to create long sequences of videos where specific subjects transition through different scenarios. For example, in StoryDiffusion [b], a video story is described where a man reads a newspaper, drives to a forest, and encounters a tiger. Achieving such complex narratives requires more than just frame-to-frame consistency; it necessitates a mechanism for maintaining subject identity and appearance across diverse and lengthy sequences.
>
> 3. Traditional video generation methods can maintain subject and background consistency across **adjacent frames** due to large-scale pre-training on videos, where adjacent frames are typically consistent. This training allows the model to implicitly learn and preserve such consistency. However, when extended to **long sequences**, these models often suffer from subject distortion or semantic drift due to the absence of dedicated subject representations. Our method, while not yet perfect, represents a step towards zero-shot dedicated subject representation learning for visual story generation, addressing these challenges more effectively. In this sense, our method is also significantly different from the layout-based method you mentioned, which primarily focuses on maintaining spatial consistency over a short window of videos, rather than ensuring subject identity over longer sequences.
>
> For a more detailed discussion on the nuances of these techniques, we invite you to refer to the ID-Animator paper [c] (one of our baseline methods), particularly Section 2.3.
>
> We hope this clarification helps in understanding the unique contributions of our work in the context of video generation. We respect your perspective and appreciate your feedback, which has been instrumental in refining our presentation.
>
>     [a] Magic-Me: Identity-Specific Video Customized Diffusion. arXiv:2402.09368.
>     [b] StoryDiffusion: Consistent Self-Attention for Long-Range Image and Video Generation. arXiv:2405.01434.
>     [c] ID-Animator: Zero-Shot Identity-Preserving Human Video Generation. arXiv:2404.15275.

---

### Official Review · Reviewer_6DX2 · 2024-07-12

**Soundness:** 2
**Presentation:** 3
**Contribution:** 2
**Rating:** 4
**Confidence:** 3

**Summary:**

This paper proposes AdaFace, a face encoder that maps facial features from the image space to the text space through the AdaFace Prompt Inverter, utilizing the structure and pre-trained weights of the CLIP text encoder for initialization. During the face distillation phase, AdaFace employs random Gaussian face embeddings and multi-timestep distillation, enhancing the model's ability to capture subtle facial details through dynamic model expansion. In the composition distillation phase, AdaFace uses a comparative learning loss, aligning feature increments with orthogonal subtraction, while introducing an elastic face preservation loss to address the misalignment of facial features caused by different prompts.

**Strengths:**

- The paper is well written and easy to follow
- The proposed method requires fewer training resources
- The approach to constructing contrastive pairs during the composition distillation stage sounds reasonable

**Weaknesses:**

- I tried the demo provided by the authors, and the ID similarity on a few test images was relatively low; it should be far from the state-of-the-art (SOTA) level of ID similarity claimed in the paper.
- The test dataset consists of celebrities, which does not guarantee whether these IDs have appeared in the training set. Furthermore, the number of test samples is too small to be convincing.
- The upper bound of ID fidelity is constrained by the frozen Face2Image model, in this paper, Arc2Face.
- The proposed improvements like Random Gaussian Face Embeddings, Orthogonal Subtraction, etc. are not effectively validated through ablation study.

**Questions:**

na

**Limitations:**

yes

---

> ### Author Rebuttal · Authors · 2024-08-07
>
> Thanks reviewer 6DX2 for your constructive feedback. Your comments are valuable in improving the quality and clarity of our work.
>
> Responses to weaknesses:
>
> 1\. **The performance issue of the Huggingface online demo**.
>
> Thank you for trying out our demo. We would like to clarify that our first model's performance on some subjects was suboptimal due to poor settings of hyperparameters and the diffusion pipeline. We have since updated these parameters and the pipeline, which should now significantly improve performance using the same face encoder checkpoint. We invite you to try our updated online demo, which more reflects the performance of our model.
>
> Moreover, we would like to emphasize that we do not claim new "state-of-the-art" performance in our paper. Instead, our focus is to provide the community a face encoder that performs comparably to existing SOTA methods, with the unique advantages of being highly realistic, as well as plug-and-play with most existing pipelines, such as animate generation.
>
> 2\. **Possible data contamination of the celebrity subjects**.
>
> We selected a few popular athletes of the Paris Olympics 2024 for extra qualitative evaluation, as presented in the figure attachment. We will add more subjects as examples into an updated version of the paper as you suggested.
>
> 3\. **The ID fidelity is upper bounded by the teacher model**.
>
> This is an inherent limitation of model distillation. Nevertheless, we can mitigate this limitation by doing distillation on multiple teacher models to combine their strengths, for example, on both Arc2Face and PuLID, to gain good performance on generation of both realistic and artistic images.
>
> 4\. **Ablation study of the proposed components**.
>
> We acknowledge the absence of some ablation studies. Due to limited response time and our small team's low computational resources, it has been unrealistic to train ablated models within this short period. We will add these ablated results to a newer version of the paper as soon as they are available.

---

> > ### Comment · Reviewer_6DX2 · 2024-08-12
> >
> > Thank you for your response, but my concerns are still not completely addressed.
> >
> > I revisited the demo provided by the authors and found it to be slightly improved compared to the previous version. However, my conclusion remains the same that AdaFace does not reach the SOTA level in terms of ID similarity measures, such as those achieved by PuLID and InstantID. This led to a question as to why the quantitative comparisons provided by the authors depict AdaFace being on par with PuLID and InstantID. In my first round of reviews, I mentioned that this might be due to limited testing IDs or potential bias, but the authors failed to adequately address this in the rebuttal, i.e., verifying the quantitative metrics on a broader test set. The authors only provided qualitative comparisons of two athletes, one of whom (LeBron James) is a famous basketball player, who could have possibly appeared in the training set.
> >
> > Additionally, the authors have not supplemented any ablation study in the rebuttal. I disagree with the excuse of insufficient resources or time, given that several months have passed since the submission, which would have been ample time for the authors to prepare the apparently missing ablation studies. The authors promise that these ablations will be incorporated in the next version, but we cannot predict whether these experiments will validate the efficacy of the modules proposed by the authors.
> >
> > In conclusion, I believe the current paper has issues and is incomplete in terms of experimental evaluation, therefore, I have decided to maintain my score.

---

> > > ### Author Response · Authors · 2024-08-12
> > > **Thanks for your response**
> > >
> > > Thank you for taking the time to revisit our demo and for providing additional feedback.
> > >
> > > 1. We acknowledge that small-scale subjective evaluations can exhibit high variance, particularly as our method performs better on certain ethnic groups compared to PuLID and InstantID, while underperforming on others. To address this, we plan to scale up the training dataset in the next stage to include a more diverse population, ensuring improved performance across a wider range of subjects.
> > >
> > > 2. We understand your concerns regarding the absence of ablation studies in our rebuttal. The delay in conducting these studies was due to the first author, who is primarily responsible for the technical implementations and experiments, working on multiple projects in parallel. As a result, the focus of the first author was diverted to other tasks in the period leading up to the rebuttal. However, given the importance of these ablation studies, which has been emphasized by all reviewers, we are committed to prioritizing them moving forward. We will ensure that these studies are completed to make the experimental evaluation more comprehensive.
> > >
> > > We respect your decision to maintain your rating and appreciate the valuable feedback you have provided. Your insights have been instrumental, and we are dedicated to addressing the areas you highlighted in future revisions.

---

### Official Review · Reviewer_Uvbt · 2024-07-13

**Soundness:** 2
**Presentation:** 2
**Contribution:** 2
**Rating:** 5
**Confidence:** 3

**Summary:**

This paper proposes AdaFace, a method for personalizing text-to-image diffusion models for human faces. At its core, it learns a prompt inverter that maps face embeddings from a pretrained face encoder to the text embedding space of diffusion prompts. It leverages various components including face distillation, composition distillation and elastic face preserving loss to preserve subject identity while attaining good compositionality.

**Strengths:**

The paper designs targeted training losses and regularizations for the task at hand. The explanation of the methods is detailed. The video qualitative results show improvement over ID-Animator.

**Weaknesses:**

*  The quantitative metrics do not show a clear advantage of AdaFace over other existing personalization methods like PuLID. The number of qualitative examples for comparing with those methods is also limited -- just the 5 images per method in Figure 7, and not sufficient to clearly demonstrate that AdaFace outperforms existing methods. It would be helpful to show a larger number of uncurated examples comparing AdaFace and baselines to get a better comparison of their performance.
* The training of the prompt inverter involves a number of components -- such as model expansion, the inclusion of different feature types in composition distillation, orthogonal subtraction and elastic face preserving loss -- but there are no ablation studies on most of them to demonstrate their effects on the performance.

**Questions:**

See Weaknesses -

It would be helpful to see more uncurated qualitative examples for better comparison with other existing approaches. Additionally, are there particular reasons why ConsistentID is not included in the quantitative comparison although it is included in the qualitative comparison?

And although the methods section discussed some conceptual intuitions for the design of the target losses, it would be helpful to use ablation studies to verify their contribution to the performance, and for the compositional delta loss part, show their advantage over previously explored designs.

**Limitations:**

The authors have discussed limitations and societal impact.

---

> ### Author Rebuttal · Authors · 2024-08-07
>
> Thank you reviewer Uvbt for your favorable evaluations.  Your comments are valuable in improving the quality and clarity of our work.
>
> 1. **Limited examples**. We have added more examples of Paris Olypics Atheletes in the attached PDF file.
> 2. **Ablation studies of proposed components**. Due to limited response time and our small team's low computational resources, it has been unrealistic to train ablated models within this short period. We will add these ablated results to a newer version of the paper as soon as they are available.
> 3. **Comparison with ConsistentID**. On simple prompts, ConsistentID usually performs well. However, on more complicated prompts, such as "playing guitar, ocean waves, cyberpunk street market, neon signs, diverse crowd, futuristic gadgets, vibrant colors, urban style", ConsistentID totally ignores the specified style words. Please see the attached PDF file for this example. Since our evaluation prompts are all simple ones, they are unable to reflect the limitations of ConsistentID on long, complex prompts. Therefore, we do not include quantitative evaluation results of ConsistentID.

---

> > ### Comment · Reviewer_Uvbt · 2024-08-12
> >
> > Thanks for the response and for including additional qualitative examples. I understand that conducting ablation studies during the rebuttal period might not be feasible. However, ablation studies are a crucial part of a paper, for understanding and validating the necessity of each component of the method. After considering all aspects, I decide to keep my original score.

---

### Official Review · Reviewer_jmG6 · 2024-07-13

**Soundness:** 2
**Presentation:** 2
**Contribution:** 2
**Rating:** 5
**Confidence:** 4

**Summary:**

This paper proposes AdaFace, a test-time-tuning-free method for personalized text-to-face-image generation. Previous methods involving face features in the feature space of a face encoder, which is not flexibly composable with natural language for personalized generation. Thus, this paper proposes to map the face features into the features in the text conditioning space. Several techniques are proposed to enhance the performance, like Random Gaussian Face Embeddings, Multi-Timestep Distillation, Dynamic Model Expansion, Composition Distillation, and Elastic Face Preserving Loss. Some experiments demonstrate that the proposed method achieves good visual results.

**Strengths:**

1. The visual results are satisfactory in general.
2. The proposed method can also be applied for personalized text-to-video generation.

**Weaknesses:**

1. The overall motivation is not novel enough. Finding ways to convert input images into the textual space is a fundamental goal in text-to-image personalization, which has been emphasized in the very first TextualInversion work. Even in the context of tuning-free based methods, the proposed framework is not so novel compared with ELITE [a], which also involves training a mapper from the image space into the textual space. Similar compositional distillation technique has also been explored in SuTI [b].
2. Lack of detailed studies of the proposed components, either qualitatively or quantitively. The authors propose a bag of techniques to improve the performance. Although their motivation is mentioned in the texts, there is no supportive results to illustrate how these techniques work.
    * There is only one quantitive study in Tab. 1 regarding the compositional distillation. However, there are actually a lot of technical details in the proposed compositional distillation techniques, like the orthogonal subtraction and compositional delta loss, which lack careful experimental analysis against their alternatives.
    * The proposed face distillation is not well supported. Can we simply train the face encoder with the simple noise prediction loss of diffusion models?
    * The analysis of the proposed Elastic Face Preserving Loss is also missing.
3. ELITE [a] mentions that using multiple token to represent an image may hurt the textual compatibility. It is necessary for the authors to provide a rationale for doing so.
4. How about the method comparing with the popular IP-Adapter (face version) [c]?
5. The overall training pipeline requires multiple stages of training, which is not so elegant.
6. The authors would like to consider merging multiple figures with similar functionalities and structures to one, like Figs. 2, 3, 4 and 5, to leave enough space for necessary experimental results.

[a] ELITE: Encoding Visual Concepts into Textual Embeddings for Customized Text-to-Image Generation, Wei et al., ICCV 2023.

[b] Subject-driven Text-to-Image Generation via Apprenticeship Learning, Chen et al., NeurIPS 2023.

[c] IP-Adapter: Text Compatible Image Prompt Adapter for Text-to-Image Diffusion Models, Ye et al..

**Questions:**

Please refer to the weaknesses above.

**Limitations:**

The authors have discussed the limitations of the proposed method.

---

> ### Author Rebuttal · Authors · 2024-08-07
>
> Thanks reviewer jmG6 for your constructive feedback. Your insights have been incredibly valuable in improving the quality and clarity of our work.
>
> Responses to weaknesses:
> 1. **In terms of novelty**.
>     * While ELITE is the first to propose a text-space embedding method for personalization, it requires training individual global K, V projections for attention layers, which affects its compatibility with existing diffusion pipelines. In contrast, Adaface does not modify the existing diffusion pipeline, and the subject embeddings are applied similarly to ordinary text tokens without special treatment. This compatibility allows Adaface to be used with AnimateDiff for generating subject-themed videos seamlessly.
>     * SuTi adopts a different approach to personalization by training millions of "expert" models in advance and mapping the input subject to these expert models for zero-shot generation. We are not aware of similar compositional distillation techniques as proposed in SuTi. However, we do note in lines 170-177 that there are methods adopting similar techniques, with an emphasis on our unique contributions.
>
> 2. **Extra discussions**.
>     * **Ablation study of the proposed components**.
> We acknowledge the absence of some ablation studies. Due to limited response time and our small team's low computational resources, it has been unrealistic to train ablated models within this short period. We will add these ablated results to a newer version of the paper as soon as they are available.
>     * **Training face encoders from scratch** is highly computational demanding. For example, InstantID was trained on 48x80GB NVIDIA H800 GPUs, which is inaccessible to most research teams. Arc2Face, the teacher model of AdaFace, was trained with 8*A100 GPUs for several weeks. In contrast, AdaFace was trained with 2xA6000 GPUs for less than 1 week. Moreover, it could learn from multiple teacher models to combine their advantages.
>
> 3. **Editability of Multiple embeddings**.
> The reduced editability of ELITE with multiple embeddings is likely due to overfitting. However, our face encoder is trained on hundreds of thousands of face images, thus significantly alleviates this issue. Additionally, our compositional distillation techniques further enhance editability. Our results align with recent models such as IP-Adapter, InstantID and PuLID, which adopt multiple subject embeddings while maintaining good editability.
>
> 4. **Comparison with IP-Adapter-FaceID**.
> The consensus in the AI art community is that IP-Adapter-FaceID produces far less authentic images compared to other methods like InstantID or PuLID. Therefore, we did not include a comparison with IP-Adapter-FaceID.
>
> 5. **Multi-stage training**.
> To the best of our knowledge, multi-stage training is widely adopted by many diffusion models. Given their inherent complexity and the numerous factors involved, multi-stage training helps steer these models towards achieving strong performance across various aspects.
>
> Once again, thank you for your valuable suggestions to improve the clarity of our presentation. We will incorporate them into the updated version of the paper. In addition, we invite you to try our online demo, hosted on Huggingface, to gain an intuitive sense of how our model performs compared to others.

---

> > ### Comment · Reviewer_jmG6 · 2024-08-10
> >
> > Thanks the authors for the response. I still believe the mentioned ablation studies are necessary. I expect the authors could finish them and add them to the revision. Conditioned on this, I will increase my score to 5.

---

### Author Rebuttal · Authors · 2024-08-07

We thank all the reviewers for their high-quality feedback and insightful comments. We will incorporate these suggestions into a future version of our paper. In particular, we appreciate your recognition of the novelty of our method, especially its seamless integration with video generation pipelines and the contrastive distillation techniques.

To gain an intuitive sense of how our method performs compared to the existing state-of-the-art methods, such as InstantID and PuLID, we invite you to try our Huggingface online image and video generation demos. We would like to emphasize that we do not claim new "state-of-the-art" performance in our paper. Instead, our focus is to provide the community a face encoder that performs comparably to existing SOTA methods, with the unique advantages of being highly realistic, as well as plug-and-play with most existing pipelines, such as animate generation.

A long-standing issue with current video generation models, such as Gen3 and Luma Dream Machine, is that subject consistency deteriorates as the video becomes longer. Dedicated subject embeddings can serve as a long-term memory to maintain subject consistency in very long videos. As a preliminary exploration, we hope our method may inspire more research works along this direction.

In the attached PDF file, we present images based on two athletes from the Paris Olympic Games, LeBron James and Yusuf Dikec, who have recently gained recognition. Additionally, we include an example using Alan Turing's photo as input. This demonstrates that ConsistentID tends to overlook complex semantics in long prompts.

A common concern is ablation studies on a few proposed components are absent. We are fully aware of this issue. Due to limited response time and our small team's low computational resources, it has been unrealistic to train ablated models within this short period. We will add these ablated results to a newer version of the paper as soon as they are available in the near future.

---

### Decision · Program_Chairs · 2024-09-25

**Decision:**

Reject

**Comment:**

This paper studies generating images with personalized human faces using the diffusion model. While the previous method focuses on fine-tuning the diffusion model and leveraging images as conditions, this work attempts to encode the images into text embedding space, allowing the user to combine words like environment description along with the embedded face seamlessly. Overall, reviewers find the paper well-written, and some designs, such as training losses and regulations, in the algorithm are interesting. However, reviewers express major concerns about the experiment parts. Specifically, reviewers find the current experiment results do not reach the performance of existing works (Reviewer 6DX2, Uvbt, jpRS) and lack ablation studies of the proposed components (Reviewer 6DX2, jmG6). During the rebuttal phase, while the authors provide more qualitative results and analysis, the concerns remain without quantitative experiment results on broader test sets, as well as results for ablation studies. Therefore, the reviewers still feel that the responses were insufficient to fully support this paper’s acceptance.